# Divergences between Language Models and Human Brains

## Abstract

Do machines and humans process language in similar ways? A recent line of research has hinted in the affirmative, demonstrating that human brain signals can be effectively predicted using the internal representations of language models (LMs). This is thought to reflect shared computational principles between LMs and human language processing. However, there are also clear differences in how LMs and humans acquire and use language, even if the final task they are performing is the same. Despite this, there is little work exploring *systematic differences* between human and machine language processing using *brain data*. To address this question, we examine the differences between LM representations and the human brain's responses to language, specifically by examining a dataset of Magnetoencephalography (MEG) responses to a written narrative. In doing so we identify three phenomena that, in prior work, LMs have been found to not capture well: emotional understanding, figurative language processing, and physical commonsense. By fine-tuning LMs on datasets related to these phenomena, we observe that fine-tuned LMs show improved alignment with human brain responses across these tasks. Our study suggests that the observed divergences between LMs and human brains may stem from LMs' inadequate representation of these specific types of knowledge[1].

## 1 Introduction

Language models (LMs) now demonstrate proficiency that may equal or even surpass human-level performance on various benchmarks involving generating contextually relevant text (Brown et al., 2020a), answering questions (Lewis et al., 2019), translating languages (Costa-jussà et al., 2022), and even tasks that necessitate reasoning and inference (Dasgupta et al., 2022). This has inspired numerous researchers to leverage LM representations to investigate and model the human brain's language system, positing that LMs might serve as a reliable proxy for human linguistic processes (Abdou, 2022). Prior studies have found that human neural activity, as reflected by neuroimaging techniques like fMRI (Jain & Huth, 2018; Toneva & Wehbe, 2019), EEG (Hale et al., 2018), MEG (Wehbe et al., 2014b), and ECoG (Goldstein et al., 2022), can be effectively predicted using representations from language models such as BERT (Devlin et al., 2018) and GPT-2 (Radford et al., 2019b). This correlation is hypothesized to stem from the shared computational objective of both LMs and the human brain: predicting subsequent words based on prior context (Schrimpf et al., 2021).

However, besides evident behavioral similarities, the extent to which LMs and human brains align functionally in language processing remains an open question. Essentially, the methods that LMs and humans use to acquire language are very different. LMs primarily learn through recognizing statistical regularities in surface-level linguistic symbols, whereas humans may rely on more structured linguistic principles. Additionally, LMs that are confined to linguistic data may fail to ground linguistic symbols in real-world contexts. This grounding is essential for humans to understand language within a broader context (Harnad, 1990; Bender & Koller, 2020; Bisk et al., 2020a). Furthermore, the contexts where LMs and humans learn language are markedly different. While humans often communicate through active inquiry, expressing needs, and scaffolding conversations

---

[1]Data and code are available at anonymized repository: `https://anonymous.4open.science/r/divergence_MEG-F647`

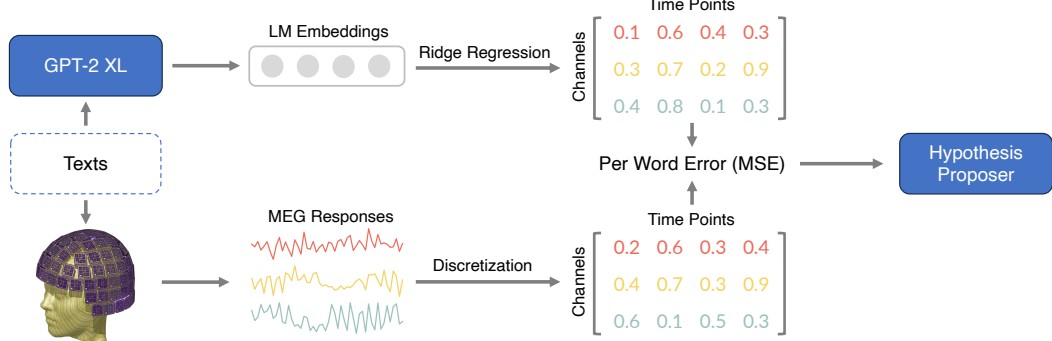

Figure 1: Schematic of our experimental approach. The LM takes as input the current word along with its preceding context to produce the current word's embedding (last hidden layer). This embedding is then used as input to a ridge regression model to predict the MEG responses associated with the current word. The Mean Squared Error (MSE) between the predicted and actual MEG responses is calculated. Finally, an LLM-based hypothesis proposer is employed to formulate natural language hypotheses explaining the divergence between the language model and the human brain.

(Kuhl, 2011), LMs are predominantly trained as passive recipients of raw text data. Consequently, LMs may struggle with comprehending social pragmatics and the nuances of words whose meanings fluctuate across different social contexts (Mahowald et al., 2023).

We present the first endeavor, to our knowledge, to systematically explore the differences between human and machine language processing using brain responses recorded by Magnetoencephalography (MEG) as participants engage in reading narratives. Our main contributions are as follows:

1. In contrast to prior studies focusing on the similarities between LMs and human brains, our research emphasizes their differences. Leveraging the high temporal resolution of MEG, we monitor the temporal progression of errors in LM predictions on a word-by-word basis (§2).

2. Explaining the prediction errors for every word is challenging due to the vast amount of text. Instead of manually formulating hypotheses, we adopt an LLM-based method that automatically proposes natural language hypotheses to explain the divergent responses between human brains and language models (§3). The top candidate explanations are related to emotion, figurative language, and physical commonsense (§4).

3. We present evidence that fine-tuning LMs on tasks associated with these three identified phenomena can align them more closely with human brain responses. This implies that the observed divergences between LMs and human brains may stem from LMs' inadequate representation of these specific types of knowledge (§5).

## 2 PREDICTIVE MEG MODEL

### 2.1 DATA PREPARATION AND PREPROCESSING

While many studies investigating the correlation between brain responses and language models utilize fMRI recordings (e.g., Caucheteux et al., 2023; Jain et al., 2020), the limitation of fMRI is its relatively low temporal resolution, which is much coarser than the time required to process individual words. Therefore, we used a MEG dataset (Wehbe et al., 2014b; Wu et al., 2022) with eight participants reading Chapter 9 of *Harry Potter and the Sorcerer's Stone*. A total of 5,176 words were sequentially displayed on the screen, with each word being exposed for a fixed duration of 500 milliseconds. In addition, we included data from four participants who read Chapter 10 of the same book, consisting of 4,475 words. This additional data was used as a held-out test set for validation[2].

---

[2]This data was obtained upon request from the authors of Wehbe et al. (2014b).

Figure 2: Pearson correlation of actual MEG responses with those predicted by the LM (evaluated on the test set). The displayed layout is a flattened representation of the helmet-shaped sensor array. Deeper reds indicate more accurate LM predictions. Language regions are effectively predicted in language processing time windows (refer to §2.3 for more details).

MEG data were collected from 306 channels at 102 cranial points, and sampled at a rate of 1 kHz. The acquired data underwent preprocessing procedures using the Signal Space Separation (SSS) method (Taulu et al., 2004) and its temporal extension, tSSS (Taulu & Simola, 2006). The signal was then time-locked with individual words and down-sampled into non-overlapping 25ms time bins. Given the typical low Signal-to-Noise Ratio (SNR) associated with MEG, we adopted a denoising technique (Ravishankar et al., 2021) that takes advantage of cross-subject correspondences to get an aggregated, denoised version of MEG responses (refer to Appendix A for more details).

## 2.2 PREDICTING MEG RESPONSES FROM LM REPRESENTATIONS

A substantial number of recent studies exploring the correlation between brain responses and LMs have employed GPT-2 (Pasquiou et al., 2022; Caucheteux et al., 2022; 2023; Toneva et al., 2022). To ensure consistency and comparability with these studies, we utilized the pre-trained GPT2-xl model with 1.5B parameters, sourced from HuggingFace's `transformers` library (Wolf et al., 2020a), as the backbone language model. For every word $w$, we provided the model with a context consisting of the preceding 99 words. We used the last hidden layer of the LM, subsequently referred to as LM embeddings, to predict the MEG responses associated with each word (Figure 1).

Building upon established research that demonstrates the capability of LM embeddings to linearly predict MEG responses (Wehbe et al., 2014b; Jain & Huth, 2018; Caucheteux & King, 2022a), we utilized a ridge regression model as the encoding model. Considering the time-correlated nature of MEG data, it was essential to maintain the temporal structure when partitioning the data for training and testing purposes (Yang et al., 2019). Therefore, we implemented a 10-fold cross-validation procedure to obtain LM predictions of MEG responses. For split $i$, we set aside one fold as the test set $L^{i,test}$ and fitted a linear ridge regression model with weight matrix $W^i$ and bias $b^i$ using the remaining folds, denoted as $L^{i,train}$. The regularization parameters were chosen via nested cross-validation. Following model training, we applied the trained weight matrix and bias to predict the brain responses from the LM outputs for the test set:

$$\hat{M}_{LM}^{i,test} = L^{i,test}\hat{W}^i + \hat{b}^i$$

Finally, the test predictions from all folds were concatenated to form the comprehensive prediction of MEG responses from the LM:

$$\hat{M}_{LM} = concat_i[\hat{M}_{LM}^{i,test}]$$

## 2.3 SPATIO-TEMPORAL PATTERNS OF CORRELATION BETWEEN LMS AND MEG

As a sanity check, we calculated the Pearson correlation between the actual MEG responses and those predicted by the language model to determine if the model can effectively predict the brain

1. He had been looking forward to learning to fly more than anything else.

2. "Of course he has," said Ron, wheeling around.

3. But Neville, nervous and jumpy and frightened of being left on the ground, pushed off hard before the whistle had touched Madam Hooch's lips.

most divergent          least divergent

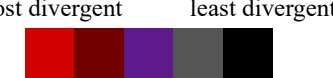

Figure 3: Sample sentences from the dataset, with colors indicating prediction error levels. Each of the five colors corresponds to a 20-percentile range of words from the entire dataset.

areas[3] and time course of language processing. As shown in Figure 2, we observe a temporal progression of accurately predicted areas after word onset. The prediction performance peaks first in the occipital lobe between 75-100ms. Given that LM embeddings encode information (e.g., word frequency) correlated to the number of letters in a word, and since MEG is sensitive to abrupt changes in visual inputs, we attribute this early peak to the initial visual perception of a word. This is followed by heightened prediction performance in the bilateral temporal lobe between 175-250ms, when we expect semantic processing to start. This observation aligns with previous research indicating that most language experiments with naturalistic stimuli reveal bilateral language representations (Wehbe et al., 2014a; Huth et al., 2016; Deniz et al., 2019; Toneva et al., 2022). Finally, between 250-375ms, the anterior temporal lobe and frontal lobe show increased prediction performance, which is likely related to further semantic processing. This sequential pattern of prediction performance replicates the spatio-temporal dynamics of language processing found in previous literature (Wehbe et al., 2014b; Toneva et al., 2022).

## 3   IDENTIFYING PHENOMENA OF INTEREST

Our primary objective is to investigate the elements of MEG responses that cannot be explained well by the LM. We work with an average of cleaned MEG responses from a group of subjects, which we anticipate should illustrate the common elements of language processing across individuals. Therefore, for words where MEG responses are not well predicted, it is likely that this marks a genuine divergence between human brains and the language model. Leveraging the high temporal resolution of MEG, we computed the Mean Squared Errors (MSEs) between actual and predicted MEG responses for each individual word on channels that demonstrated statistically significant correlation[4].

### 3.1   AUTOMATICALLY DISCOVERING DIFFERENCES BETWEEN BRAIN AND LM PREDICTIONS

Given the vast amount of text, manual pattern discovery becomes challenging (refer to Figure 3 for sample sentences). To discover subtle differences between MEG responses and LM predictions, we used a method that automatically describes differences between text corpora using proposer and verifier LMs (Zhong et al., 2023). This system consists of first prompting an LLM (GPT-3; Brown et al. (2020b)) with a number of samples from two corpora $(D_0, D_1)$ to generate many hypotheses on how the first corpus differs from the second, and then using a fine-tuned validator model (FLAN-T5-XXL; Chung et al. (2022)) to validate how often each proposed hypothesis is true based on pairs from each corpus sampled from a held-out set. Specifically, the verifier is presented with a prompt containing two sentences from $D_0$ and $D_1$, and asked whether or not the hypothesis is true, and this is repeated across the development set for each hypothesis. We note that although hypotheses proposed by GPT-3 may not all be well-supported, especially given that not all sentences fit in its context window, this was accounted for by the method. Namely, verifying hypotheses after they are proposed is a much easier problem.

---

[3]These areas include the inferior frontal gyrus, superior temporal gyrus, certain sections of the middle temporal gyrus, and angular gyrus (Blank et al., 2016; Rogalsky et al., 2015; Sahin et al., 2009; Brennan & Pylkkänen, 2012; Friederici, 2002; Visser et al., 2010; Rogalsky & Hickok, 2009).

[4]See Appendix B for the number of significant channels across time.

Table 1: Top 10 hypotheses found by the hypothesis proposer from Chapter 9, ranked by validity

| Hypothesis | Theme | Validity | p-value |
|---|---|---|---|
| includes a reference to nature or the outdoors | Physical | 0.2355 | 0.0172 |
| employs a reference to a mythological creature or figure | Physical | 0.2337 | 0.0249 |
| contain a reference to a character's fear or anxiety | Emotion | 0.1847 | 0.0387 |
| employ figurative language or metaphor | Figurative | 0.1811 | 0.0678 |
| uses a rhetorical question | - | 0.1757 | 0.0925 |
| contains a reference to a magical object or creature | Physical | 0.1641 | 0.0855 |
| contain figurative language, like metaphors, similes, and personification | Figurative | 0.1641 | 0.0855 |
| contains metaphors or figurative language | Figurative | 0.1525 | 0.0359 |
| mention a character's struggle to overcome a challenge | Emotion | 0.1356 | 0.0462 |
| contain rhetorical questions | - | 0.1285 | 0.1487 |

This process of hypothesis proposal and verification was repeated across 3 cross-validation folds. We used entire sentences as input to the proposer. We identified the top 100 words where the brain responses were most accurately predicted by the LM, as opposed to the 100 least accurately predicted words. The sentences encompassing these words were labeled as $D_0$ and $D_1$. We then ran the pipeline on these sentences.

The top ten hypotheses ranked by validity[5] are listed in Table 2. It should be noted that due to the dataset's relatively small size, some of the p-values may not be statistically significant. To validate our findings, we extended our analysis to Chapter 10 of the same book (Appendix D). This replication produced slightly varied but fundamentally similar topics to those discovered in Chapter 9. Importantly, these hypotheses resonate with conclusions drawn in prior research, as detailed in §4. We identify two primary differences between the language model and the human brain: the processing of **emotion** and **figurative language**. Other hypotheses also highlight aspects related to characters, magical creatures, and nature – we extend our analysis by condensing these into a single hypothesis encompassing **physical commonsense**.

## 4    Selected Phenomena

To comprehend domains like emotion, figurative language, and physical commonsense, humans use a broad spectrum of contextual knowledge. We briefly discuss the insights and challenges highlighted in the existing neuropsychological and NLP literature regarding these domains.

### 4.1    Emotions and Social Intelligence

Emotions extend beyond introspection; they encompass predicting the feelings of others. Consequently, a comprehensive understanding of emotions involves social and emotional intelligence regarding others (Salovey & Mayer, 1990). Under this view, emotions are intrinsic to the human experience and pervasively interact with other mental facilities, including language (Satpute & Lindquist, 2021). Neuropsychologically, research on social cognition has identified a network of brain regions that support understanding other people's intentions, actions, and emotions (Saxe et al., 2006).

Within NLP, creating agents with social and emotional intelligence has been a longstanding goal (Gunning, 2018; Paiva et al., 2021). However, at present, LLMs still fall behind human abilities for inferring the mental states and emotions of others ("theory-of-mind" tasks) (Sap et al., 2022).

### 4.2    Figurative Language

Figurative language, often expressed through metaphors, similes, irony, and sarcasm, conveys meanings beyond the literal sense (Shutova, 2011). Neuropsychologically, the precise locus for processing figurative language remains debated, in part because of the difficulty of designing experiments that correctly match between metaphors and control sentences, and take into account aspects such as

---

[5] Validity measures the difference in certainty that the hypothesis is true between the two corpora, see Zhong et al. (2023) for more details.

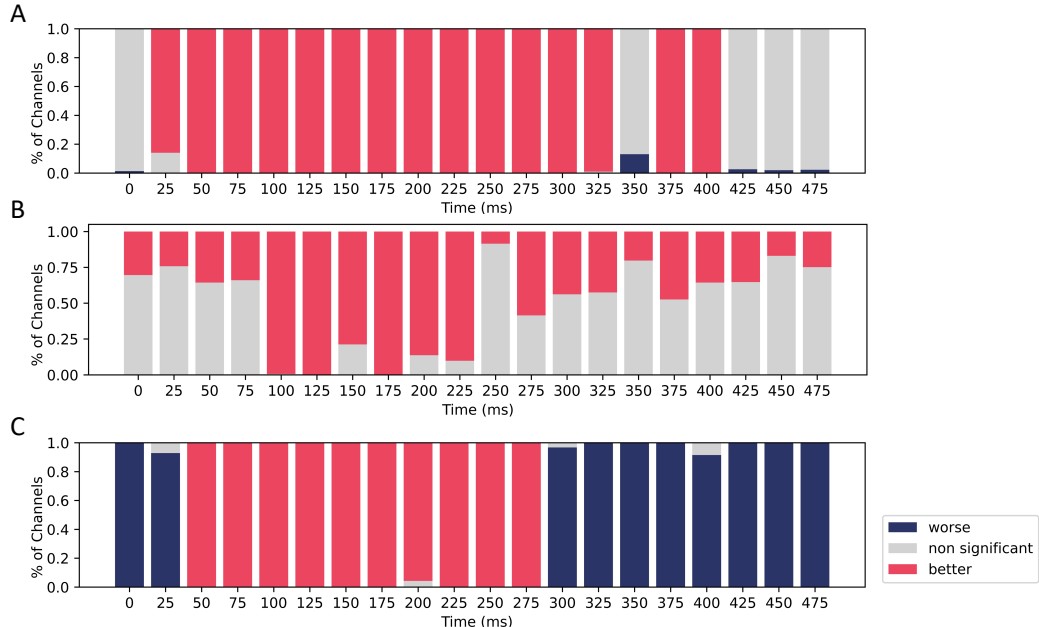

Figure 4: Performance comparison of the base model with models finetuned on (A) emotion, (B) figurative, and (C) physical datasets. Each panel's y-axis shows the percentage of channels in the finetuned model with better, worse, or non-significantly different performance compared to the base model. Finetuned models outperform the base model during language processing time windows. Refer to Appendix H for a detailed view of each channel plotted.

familiarity, difficulty, and metaphor types (Cardillo et al., 2010; Klooster et al., 2020). Nonetheless, there is growing evidence indicating a network within the left hemisphere dedicated to figurative language processing (Citron, 2020), which is consistent with the general left-hemispheric localization for language processing.

Within NLP, figurative language has not been a widely-investigated topic, in part because it relies on a wide range of cultural and contextual knowledge that is not directly carried by language. As such, language models currently underperform humans in both the interpretation and generation of figurative language (Chakrabarty et al., 2022; Liu et al., 2022) and the correct represention of idiomatic phrases (Dankers et al.; Liu & Neubig, 2022).

## 4.3 PHYSICAL COMMONSENSE

Physical commonsense refers to knowledge about the physical properties of everyday objects and physical phenomena (Forbes et al., 2019; Bisk et al., 2020b). Neuropsychologically, language is not the primary channel through which humans acquire commonsense physical knowledge. Instead, humans typically rely on sensory inputs and interactions with their environment (Baillargeon, 1994). Notably, the category of a physical object affects which brain regions are recruited when interacting with that object. For example, interacting with people activates the theory of mind areas (Saxe et al., 2006), the visual face areas (Sergent et al., 1992; Kanwisher et al., 1997), and body areas (Downing et al., 2001), while interacting with corridors while navigating will recruit the visual place areas (Epstein & Kanwisher, 1998) and spatial navigation areas. Interestingly, reading about objects has been shown to activate the visual regions that are recruited when interacting with these objects (Wehbe et al., 2014a; Huth et al., 2016).

Within NLP, acquiring physical commonsense knowledge poses a notable challenge for language models. While these models can potentially learn representations capturing specific physical properties of the world, such as an object's color or a game board's state (Abdou et al., 2021; Li et al.,

2023), it remains unclear whether text-based representations can truly capture the richness and complexity of physical commonsense as exhibited by humans (Forbes et al., 2019; Bisk et al., 2020b).

# 5 IMPROVING BRAIN ALIGNMENT VIA FINE-TUNING

We hypothesize that the LM may not capture the three language phenomena with sufficient expressiveness, hindering its ability to predict associated brain responses. Drawing inspiration from Aw & Toneva (2023), where fine-tuning on a narrative dataset enhanced brain alignment, especially for references to story characters, we fine-tuned the GPT-2 XL model on datasets specific to each of these phenomena to see if targeted fine-tuning could enhance the model's alignment with brain activity.

Furthermore, we examined whether domain-specific fine-tuning would specifically bolster the model's capability in predicting MEG responses associated with words from that domain, as compared to words outside that domain. To this end, we recruited three raters to annotate Chapter 9 of *Harry Potter* across the three domains. We release these annotations as a resource for the dataset to facilitate further analysis. Details on the annotation process can be found in Appendix E. Examples of each phenomenon within the *Harry Potter* text can be found in Appendix F.

## 5.1 DATASETS

**Emotion** We study emotion using the Social IQa dataset (Sap et al., 2019). This dataset contains questions about peoples' feelings and motivations in a given situation. Although some questions focus more on social norms than emotion, the dataset provides detailed scenarios and contains some emotional narratives, which may match with situations found in fiction.

**Figurative Language** We study figurative language using the Fig-QA dataset (Liu et al., 2022), which contains inferences based on figurative phrases. These phrases were written by crowd workers, who were given instructions to create creative yet clear metaphors.

**Physical Commonsense** We study physical commonsense using the PiQA dataset (Bisk et al., 2020b). This dataset contains goal-driven questions based on everyday situations. These questions were taken from the website instructables.com, where people share DIY project instructions.

We also provide examples from each dataset in Table 2.

Table 2: Datasets for Fine-Tuning with Sample Questions and Answers (Correct Answer in Bold)

| Dataset | Type | Num train | Num options | Sample question | Sample answers |
|---------|------|-----------|-------------|-----------------|----------------|
| Social IQa | Emotion | 33.4k | 3 | Sydney had so much pent up emotion, they burst into tears at work. How would Sydney feel afterwards? | 1. affected 2. **like they released their tension** 3. worse |
| Fig-QA | Figurative | 9.6k | 2 | Her word had the strength of titanium. | 1. **her promises can be believed.** 2. her promises cannot be trusted |
| PiQA | Physical | 16.1k | 2 | When boiling butter, when it's ready, you can | 1. Pour it onto a plate 2. **Pour it into a jar** |

## 5.2 FINETUNING SETUP

In order to keep the architecture of fine-tuned models consistent with the base model, we format the multiple choice task as $N$ language modeling tasks, where $N$ is the number of options. Specifically, for the combined context and question $x$, we directly concatenate each possible multiple-choice answer $\{y_1, ..., y_N\}$ to $x$ to form $N$ different sentences. After passing the concatenated sequences through the model, we sum the logits of all tokens corresponding to each multiple-choice option to obtain a score proportional to its log like-lihood. These scores are then gathered into a size (1,

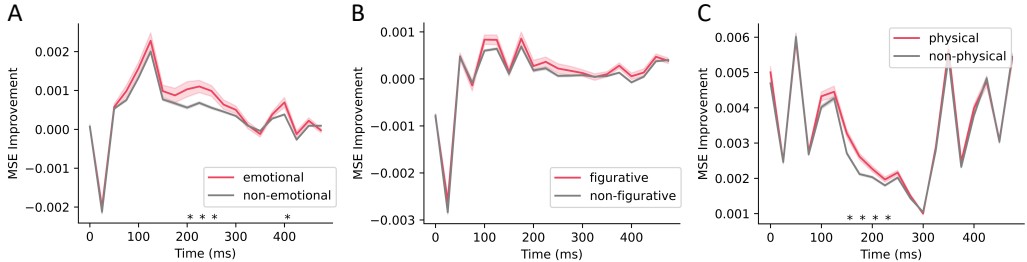

Figure 5: Comparison of improved MSE between (A) emotional, (B) figurative, and (C) physical words and those outside each category. Positive values denote lower MSEs in the fine-tuned model. Shaded region indicates standard error. Asterisks denote time points with significant differences between the two groups (Student's t-test with FDR correction, p=0.05).

$N$) tensor, and cross-entropy loss relative to the correct multiple choice answer is used to train the model. Further details on the fine-tuning setup can be found in Appendix C.

### 5.3 COMPARING FINE-TUNED MODELS WITH THE BASE MODEL

We compare the Pearson correlation between actual brain data and the predictions from the base and the fine-tuned model to compare the two models. To identify channels with statistically significant differences, we calculated empirical p-values by contrasting the true correlation value with 10,000 simulated ones obtained by permuting the brain data. Details can be found in Appendix G.

**Fine-tuned models are better aligned with the brain on all three tasks.** As illustrated in Figure 4A, the model fine-tuned on the emotion dataset exceeds the base model in performance across the majority of channels within the 50ms to 300ms time interval post word onset. Notably, this interval corresponds to the language processing time windows, as identified in §2.3. Although the fine-tuned figurative model does not significantly outperform the base model in the early and late stages, it aligns better with brain activity in the majority of channels during the 100ms-225ms interval post word onset (Figure 4B). In a similar vein, the fine-tuned physical model exceeds the base model's performance in almost all channels during the 50-275ms interval post word onset (Figure 4C). However, interestingly, almost all channels are worse than the base model outside this interval. This time selectivity indicates that the improvements of the fine-tuned model are likely tailored towards linguistic comprehension rather than broader brain functionalities.

**Fine-tuning improves alignment more for words annotated with that category.** We compared the reduction in prediction error for words annotated within each category and words outside each category by computing the difference in MSE between the model fine-tuned on the corresponding task and the base model. As demonstrated in Figure 5A, prediction errors for emotion words exhibit a significant reduction compared to non-emotion words 200-275ms post word onset. Figurative words also seem to generally yield a greater reduction in MSE than non-figurative words though we don't observe any significant time window (Figure 5B). Additionally, there is a significant improvement in MSE for physical words over non-physical words 150-225ms post word onset (Figure 5C).

**Improvements are not related to increased language-modeling ability.** Prior work has found that LMs with lower perplexity can better predict brain activity (Schrimpf et al., 2021). Therefore, one confounding factor is that the additional fine-tuning may have improved the language model's ability to perform the LM task in general, leading to improved alignment. To rule out this possibility, we performed 3-fold cross-validation on *Harry Potter and the Sorcerer's Stone*, excluding Chapters 9 and 10, which were used as data in this study. We trained the base model, as well as the finetuned emotion and figurative models, on the train set in each fold with the language modeling objective, and found that the final average losses on the test sets were similar (See Appendix I for details).

## 6    RELATED WORK

Numerous studies have found that LM hidden states can linearly map onto human brain responses to speech and text measured by MEG, EEG, and fMRI (Wehbe et al., 2014b; Hale et al., 2018; Jain & Huth, 2018; Abnar et al., 2019; Jat et al., 2019; Gauthier & Levy, 2019; Toneva & Wehbe, 2019; Caucheteux & King, 2022a; Jain et al., 2020; Toneva et al., 2022; Aw & Toneva, 2022; Oota et al., 2023; Sun et al., 2023;?; Oota et al., 2022).

At a more foundational level, studies have identified shared computational principles between LMs and human brains. Evidence suggests that both human brains and LMs are perpetually engaged in predicting the subsequent word (Schrimpf et al., 2021). LM surprisal is found to be positively correlated with brain activation, reaching its peak approximately 400 ms post word onset (Goldstein et al., 2022). This aligns well with N400, which denotes a decline in brain activation upon encountering unexpected words around 400 ms after word onset (Lau et al., 2009; Parviz et al., 2011; Halgren et al., 2002). Moreover, LM representations can predict the hierarchy of brain responses (Caucheteux & King, 2022b; Caucheteux et al., 2023). Despite this, Antonello & Huth (2022) have pointed out that a high correlation between brain activity and LMs does not necessarily imply that they operate under similar computational principles.

We not only observe this LM-brain alignment but can also actively intervene in it. Research has demonstrated that the alignment between LMs and human brains can be improved by task-specific fine-tuning. A notable instance is the study by Schwartz et al. (2019), where the fine-tuning of BERT using both fMRI and MEG signals enhanced its ability to predict fMRI responses. Importantly, this improvement was not participant-specific and could be transferred to hold-out individuals. Another study (Aw & Toneva, 2023) showed that task-oriented fine-tuning, particularly for narrative summarization, also facilitated better alignment with brain activity. Furthermore, altering the architecture of BERT such that it aligns better with the brain improves its performance on downstream NLP tasks (Toneva & Wehbe, 2019). These findings suggest a potentially symbiotic relationship between enhancing task performance in LMs and boosting their alignment with brain responses.

## 7    CONCLUSIONS, LIMITATIONS, AND FUTURE WORK

We explore a critical question connecting language models with human neural activity: How do LMs differ from human brains in processing language? We employed an LLM-based approach to automatically propose hypotheses explaining why human brains and LMs diverge, and test these hypotheses through fine-tuning language models on datasets related to these hypotheses. Emotion, figurative language, and physical commonsense emerged as the three dominant themes. After fine-tuning a base model on datasets related to these themes, we observed an improved alignment between LM predictions and human brain responses in language processing time windows. We use GPT-2 XL as the base model for these experiments in order to align with results in the previous literature, but we note that our methods can easily be extended to more recent language models, such as Llama-2 (Touvron et al., 2023).

We notice that the dataset we used for fine-tuning may present a different composition of physical entities compared to the *Harry Potter* chapters, which often feature magical objects (e.g., broomstick), fantasy creatures (e.g., Peeves), and character names. As a result, models fine-tuned on existing physical entity datasets might still not grasp certain information that causes the LM and human brain to have divergent responses.

Our study reveals varying degrees of improved alignment in models fine-tuned on different tasks. This variation may arise because fine-tuning within the language modality alone is insufficient for fully aligning a language model's understanding with human experiences. Incorporating additional modalities, such as visual and motor information, could be essential for capturing a broader spectrum of human knowledge. In future research, it would be beneficial to delve into whether the alignment can be enhanced by fine-tuning LMs across multiple modalities. This could offer insights into not only enhancing LM-brain alignment but also guiding the future design and evolution of LMs.

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

## A  MEG Pre-processing

Because of the typical low Signal-to-Noise Ratio (SNR) associated with MEG, we adopted a denoising technique (Ravishankar et al., 2021) that takes advantage of cross-subject correspondences to get an aggregated, denoised version of MEG responses.

Specifically, this process involves modeling the MEG responses $M_t$ of subject $t$ as a linear function of the MEG responses $M_s$ from a source subject $s$:

$$\hat{M}_{t \leftarrow s} = \hat{W}_{t \leftarrow s} M_s + \hat{b}_{t \leftarrow s}$$

We estimated the target subject's MEG responses from all other subjects:

$$\hat{M}_t = \frac{1}{N-1} \sum_{s \in S, s \neq t} \hat{M}_{t \leftarrow s}$$

where $S$ is the set of subjects and $N$ is the number of subjects. These individual estimates are then aggregated to generate a denoised version of MEG responses:

$$\hat{M} = \frac{1}{N} \sum_{s \in S} \hat{M}_t$$

## B  Language Channels

We identified channels where the LM prediction has a statistically significant correlation with actual MEG responses for each time window. This resulted in fluctuating counts of significant language channels over time, as depicted in Figure 6.

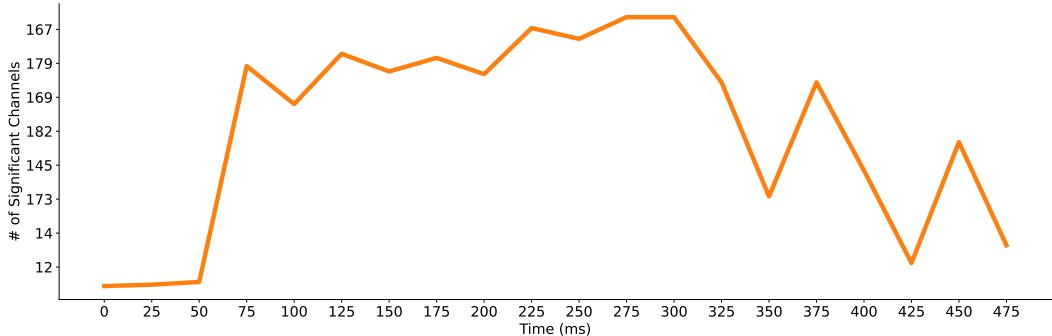

Figure 6: Number of significant language channels as a function of time. The number initially rises, remains consistent for a period, and then decreases as time progresses.

## C  FINE-TUNING DETAILS

### C.1  COMPUTATIONAL DETAILS

The model we chose to examine was GPT-2 XL, an autoregressive transformer-based model with 1.5B parameters (Radford et al., 2019a). We used the implementation in the HuggingFace library (Wolf et al., 2020b). Models were trained separately on each of the three test sets in subsection 5.1 on 4 A6000 GPUs with 16-bit quantization and a batch size of 1 per GPU. Deepspeed with ZeRo stage 2 optimization was used in order to parallelize training (Rasley et al., 2020). The Adam optimizer was used with a learning rate of 1e-5, betas of $(0.9, 0.999)$, epsilon of 1e-8, and no weight decay. Models were trained with early stopping with a patience of 3 (Kingma & Ba, 2017).

### C.2  MULTIPLE-CHOICE TRAINING

Let $x_i$ represent the concatenation of the context, if applicable, and the question. Then for each answer choice $y_i$, we concatenate it with the question and context, and feed it to the model to obtain a sequence of logits.

$$\ell_i = \text{Model}(x_i \oplus y_i)$$

Then we sum the logits corresponding to the sequence, where $t \in [1, T]$ represents the total length of $x_i \oplus y_i$.

$$\text{score}_i = \sum_{t=1}^{T} \ell_{i,t}$$

Finally, we take the cross-entropy loss of these values relative to a one-hot encoding of the correct option, where $t_i = 1$ if option $i$ is correct, or else 0.

$$P_i = \frac{\exp(\text{logit}_i)}{\sum_{j=1}^{N} \exp(\text{logit}_j)}$$

$$L = -\sum_{i=1}^{N} t_i \log(P_i)$$

### C.2.1  PERFORMANCE ON MULTIPLE-CHOICE DATASETS

We note that performance of the final model may not approach that of GPT-2 XL finetuned with an output size of $N$ denoting each option, as we keep the output dimension the same as the size of the

vocabulary. However, we report the final accuracy achieved by each model on the original datasets here.

| Dataset | Best epoch | Accuracy (%) | Baseline (random) accuracy |
|---------|-----------|--------------|----------------------------|
| Social IQa | 4 | 54.86% | 33.33% |
| Fig-QA | 1 | 85.1% | 50.00% |
| PiQA | 1 | 73.88% | 50.00% |

Table 3: Summary of model performance on common-sense related datasets.

## D HYPOTHESES ON EXTENDED DATA

We also applied the hypothesis proposer to Chapter 10 of the Harry Potter series, with the top 10 hypotheses listed in Table 4. Notably, the topics identified showed a slight variation yet maintained a resemblance to those discovered in Chapter 9.

## E ANNOTATIONS

To decide which category a word belongs to, we employed three raters who used binary coding to indicate if a word belonged to the target category. The consistency among raters was evaluated using Krippendorff's alpha. Their consistency was 0.54 for emotion, 0.44 for figurative, and 0.87 for physical. Finally, if at least two out of the three people annotated a word as fitting a category, we counted it as belonging to that category.

### E.1 ANNOTATION GUIDELINES

#### E.1.1 EMOTION

- Include words that depict the emotions of characters, primarily adjectives and adverbs.
- Exclude words that suggest emotions indirectly. For instance, "slam the door" shouldn't be annotated for emotion.

#### E.1.2 FIGURATIVE LANGUAGE

- Identify words that have meanings extending beyond their literal interpretations.
- Annotate similes: comparisons between two unlike entities using "like" or "as". E.g., "I'm free as a bird."
- Annotate metaphors: direct comparisons made without using "like" or "as". For instance, "He gave a talk following mine" exemplifies the "time is space" metaphor.

Table 4: Top 10 hypotheses found by the hypothesis proposer from Chapter 10, ranked by validity

| Hypothesis | Theme | Validity | p-value |
|------------|-------|----------|---------|
| contain figurative language | Figurative | 0.2934 | 0.0072 |
| contain references to the unknown | - | 0.2410 | 0.0312 |
| contain phrases related to the supernatural | Physical | 0.2131 | 0.0121 |
| include references to magic or fantasy elements | - | 0.2107 | 0.0392 |
| contain references to the supernatural | Physical | 0.1967 | 0.0159 |
| contain words or phrases with double meanings | Figurative | 0.1951 | 0.0568 |
| contain references to reward or punishment | Emotion | 0.1795 | 0.0373 |
| contain references to the mysterious | - | 0.1787 | 0.0521 |
| describe events with suspenseful or exciting tones | Emotion | 0.1770 | 0.0743 |
| contain unexpected or unusual words | - | 0.1746 | 0.0897 |

- Annotate personification, where non-human entities are endowed with human characteristics. E.g., "The sun smiled down on us."

- Annotate hyperbole: deliberate over-exaggeration for emphasis or effect. E.g., "I am starved to death."

- Annotate allusions: subtle references to well-known historical, cultural, or literary figures, places, or events. These presuppose the audience's prior knowledge for full understanding. An example would be, "His mistake wasn't as grave as chopping down a cherry tree."

### E.1.3 PHYSICAL COMMONSENSE

- Annotate words referring to tangible entities, such as characters (people) and physical objects.

- Do not annotate words that represent concrete ideas but lack physical substance, like "laughter".

- Pronouns should also be excluded.

## F EXAMPLES OF PHENOMENA IN HARRY POTTER

We give some examples of the three phenomena in the dataset according to the annotations. Words of that category are marked in bold.

### F.1 EMOTION

- Harry had never believed he would meet a boy he **hated** more than Dudley.

- Hermione Granger was almost as **nervous** about flying as Neville was.

- But Neville, **nervous** and **jumpy** and **frightened** of being left on the ground, pushed off hard before the whistle had touched Madam Hooch's lips.

### F.2 FIGURATIVE LANGUAGE

- His broomstick was still rising higher and higher, and started to **drift lazily** toward the forbidden forest and out of sight.

- "Ooh, **sticking up** for Longbottom?" said Pansy Parkinson, a **hard-faced** Slytherin girl.

- His heart **sank** faster than he'd just dived.

### F.3 PHYSICAL COMMONSENSE

- Up the **front steps**, up the **marble staircase** inside, and still **Professor McGonagall** didn't say a word to him.

- **Ron** had a piece of **steak** and **kidney pie** halfway to his **mouth**, but he'd forgotten all about it.

- They pulled on their **bathrobes**, picked up their **wands**, and crept across the **tower room**, down the **spiral staircase**, and into the **Gryffindor common room**.

## G ALGORITHM FOR PERMUTATION TEST

To identify channels on which the performance of the fine-tuned model and the base model has statistically significant differences, we calculated empirical p-values by contrasting the true correlation value with 10,000 simulated ones obtained by permuting the brain data as shown in Algorithm 1. Given that we are assessing multiple hypotheses simultaneously, we also used the Benjamini-Hochberg False Discovery Rate (FDR) (Benjamini & Hochberg, 1995) to correct for multiple comparisons, at level $\alpha = 0.05$.

---

**Algorithm 1** Permutation test (for one channel, one time window)

---

**Input:** Brain data $D$, Prediction from base model $P_1$, Prediction from fine-tuned model $P_2$.

    $D$, $P_1$, and $P_2$ are all of size $(1, N)$, where $N$ is the number of words in the dataset.

**Output:** `pvalue`

    $X = \mathrm{corr}(D, P1) - \mathrm{corr}(D, P2)$                     ▷ Pearson correlation coefficient

    `Counter` $= 0$

    **for** $i$ in 10,000 **do**

        $D_i = \mathrm{permute}(D)$                    ▷ Random permutation across words

        $X_i = \mathrm{corr}(Di, P1) - \mathrm{corr}(Di, P2)$

        **if** $X_i > X$ **then**

            `Counter` $=$ `Counter` $+ 1$

        **end if**

    **end for**

    Compute `pvalue` $= \frac{\texttt{Counter}+1}{10,000+1}$                  ▷ Empirical p value

---

## H COMPARISON BETWEEN FINE-TUNED MODELS AND THE BASE MODEL

We provide a detailed view of the Pearson correlation of the base model and models finetuned on emotion (Figure 7), figurative (Figure 8), and physical commonsense (Figure 9) datasets with each channel plotted.

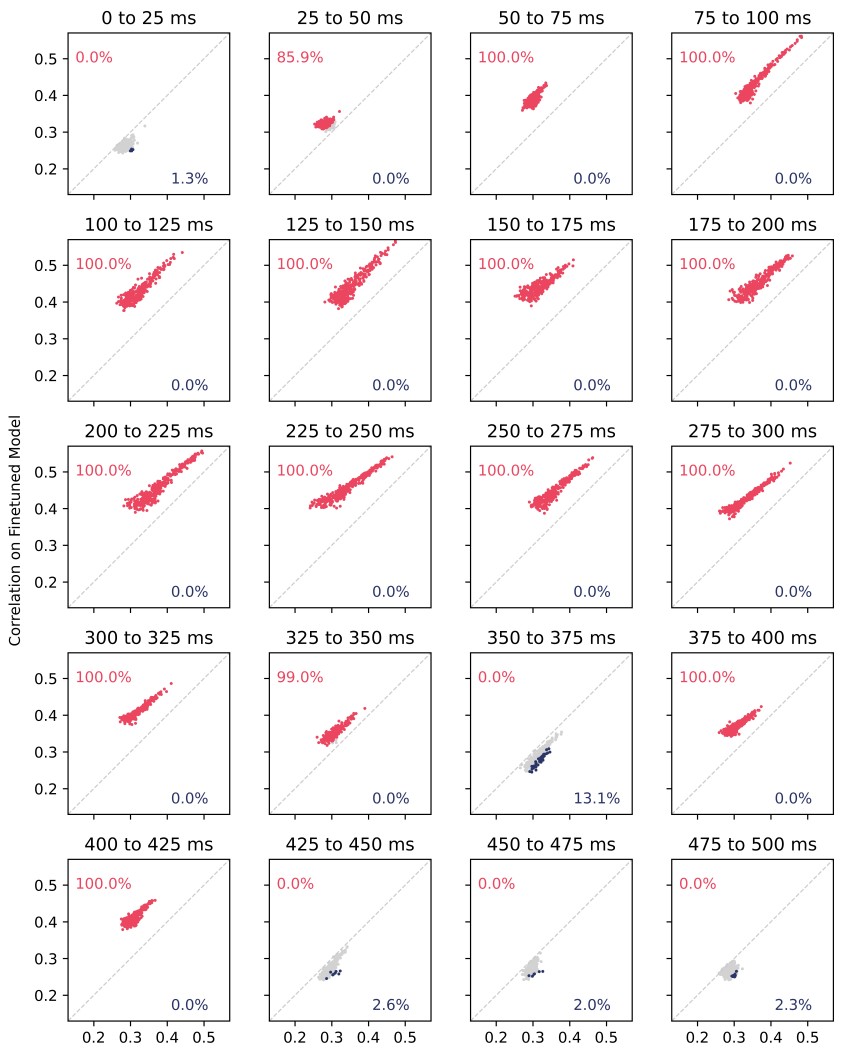

Figure 7: Performance evaluation of the model fine-tuned on the Social IQa (emotion) dataset versus the base model using Pearson correlation. Each dot represents a MEG channel. Red channels indicate better predictions by the fine-tuned model, blue channels indicate better predictions by the base model, and gray dots denote non-significant differences. The fine-tuned model outperforms the base model in predicting most channels during language processing time windows.

## I   CROSS-VALIDATION ON LANGUAGE MODELLING TASK

We perform 3-fold cross-validation on the remaining chapters of the *Harry Potter* book (excluding chapters 9 and 10), where we randomly shuffle paragraphs and assign to train:validation:test sets respectively 77%, 16.5%, and 16.5% of the data. Paragraphs that exceeded the context length were excluded. Both the base gpt-2 xl model as well as each model finetuned on the three domains were trained to predict the next word for 3 epochs, with the same hyperparameters used in Appendix C. Results on the test set for each fold are listed below. The average negative-log-likelihood loss per token at the end of training is reported in Table 5.

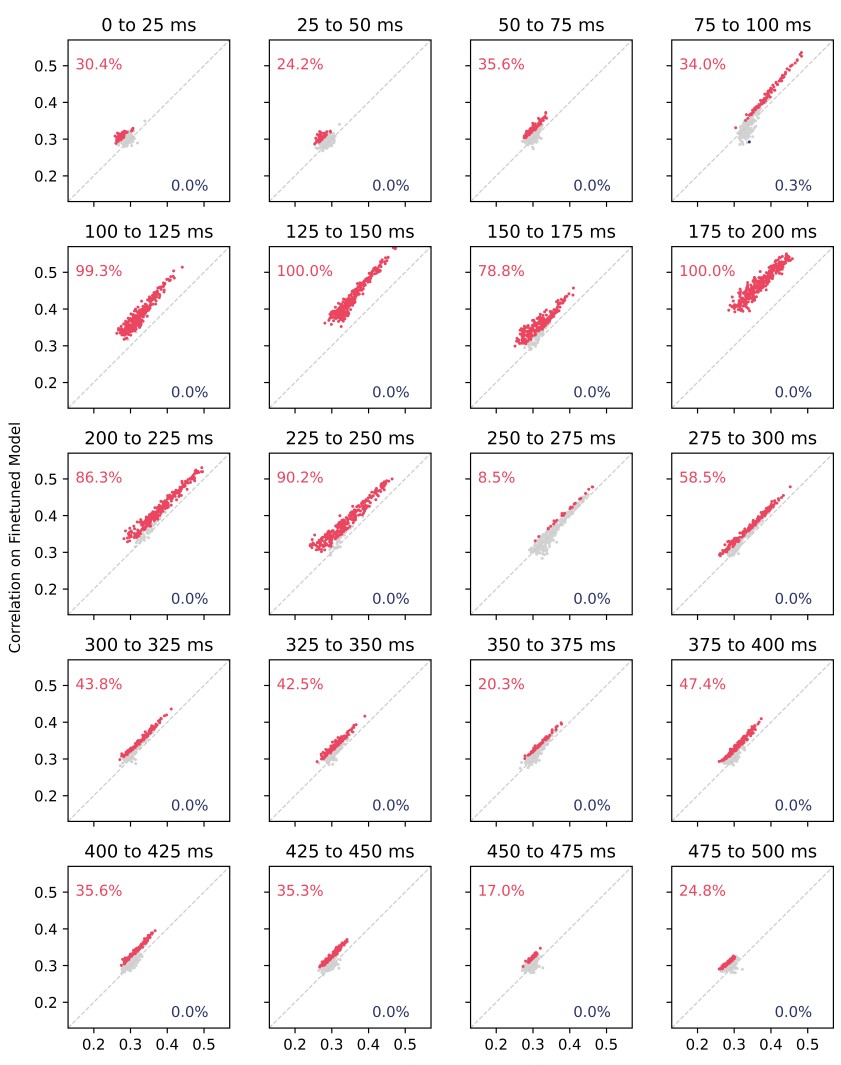

Figure 8: Performance evaluation of the model fine-tuned on the Fig-QA (figurative) dataset versus the base model using Pearson correlation. Each dot represents a MEG channel. Red channels indicate better predictions by the fine-tuned model, blue channels indicate better predictions by the base model, and gray dots denote non-significant differences. The fine-tuned model outperforms the base model in predicting most channels during language processing time windows.

| Model | Avg. Loss (%) ± St.dev | Fold 1 Loss | Fold 2 Loss | Fold 3 Loss |
|---|---|---|---|---|
| Base | 0.08795 ± 0.01707 | 0.09794 | 0.06391 | 0.1020 |
| Emotion | 0.08613 ± 0.03011 | 0.1119 | 0.1026 | 0.04388 |
| Figurative | 0.06651 ± 0.02584 | 0.09472 | 0.07252 | 0.03229 |

Table 5: Summary of language-modeling loss across cross-validation folds for models on the remaining chapters of *Harry Potter*.

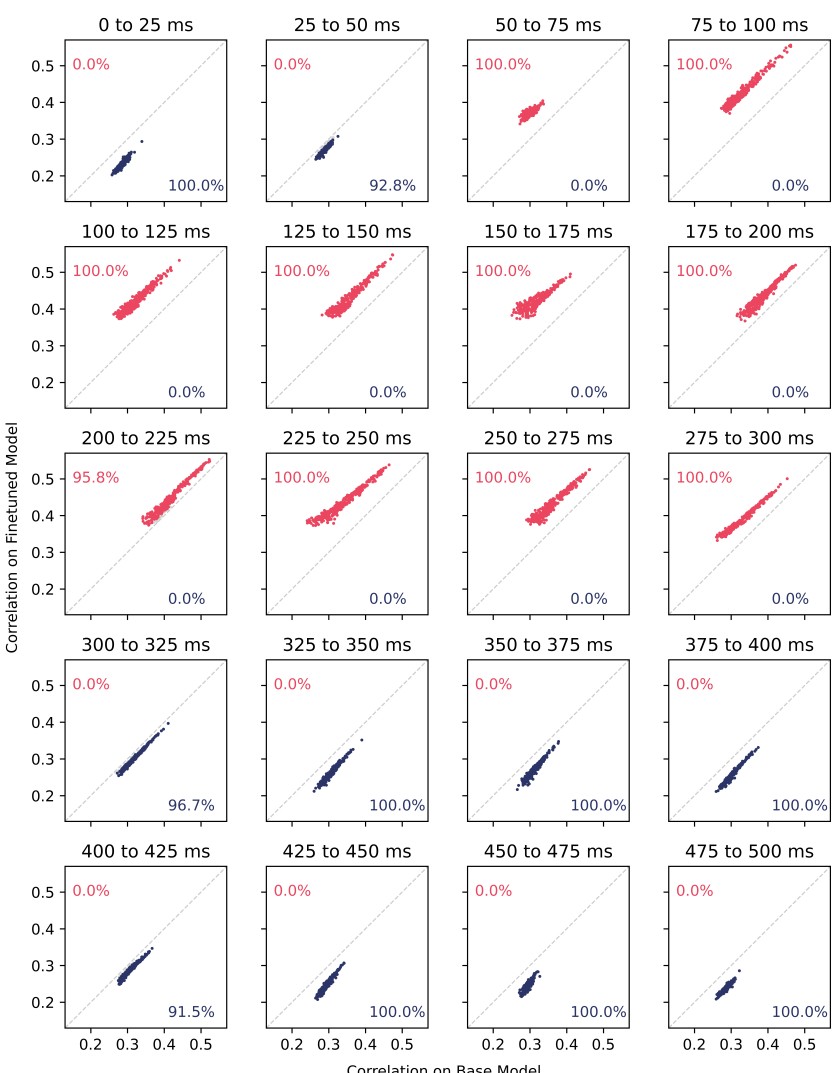

Figure 9: Performance evaluation of the model fine-tuned on the PiQA (physical) dataset versus the base model using Pearson correlation. Each dot represents a MEG channel. Red channels indicate better predictions by the fine-tuned model, blue channels indicate better predictions by the base model, and gray dots denote non-significant differences. The fine-tuned model outperforms the base model in predicting most channels during language processing time windows.

