# OpenReview forum: "Discovering Divergences between Language Models and Human Brains"
_ICLR.cc/2024/Conference — Submitted to ICLR 2024_

### Official Review · Reviewer_BQhU · 2023-10-31

**Soundness:** 2 fair
**Presentation:** 3 good
**Contribution:** 2 fair
**Rating:** 3
**Confidence:** 4

**Summary:**

The authors predict MEG responses to words from a visually presented narrative story (from Harry Potter) using the activations from the last layer of GPT-2. They then selected the 100 words that were best- and worst-predicted by the model. The sentences from these words were then fed into a second proposer-verifier LM to try and generate hypotheses for how these differ. The authors took the hypotheses and gave a subset of these hypotheses labels (“physical”, “figurative”, “emotion”) and then attempted to fine tune the base LM on three tasks that correspond to these labels. They report increased prediction accuracy for the models fine-tuned on emotion and figurative tasks, but not the physical task.

**Strengths:**

The overall approach is interesting. The authors attempt to learn in a data-driven way stimuli that are poorly predicted by the model and then try to leverage these insights to improve the model. This approach seems relatively novel and potentially promising. The writing is generally clear and the approach was well motivated.

**Weaknesses:**

I don’t think the statistical procedure used to establish significance is sufficient and it is not described in enough detail to evaluate it. In Appendix F, the authors say they permute the “data” using “random permutation”. What is the data matrix? Are they computing timepoints? Computing timepoints is not valid as this destroys the temporal structure of the data. In addition, the null hypothesis being tested is that there is no relationship between the response and either of the two predictors, which is not the correct null. The correct null is that there is no difference in prediction between the two predictors (but both predictors explain real variance nonetheless). The authors need a valid and clearly described statistical procedure to establish their main finding.

In the main figure of the paper, the authors should report a meaningful measure of effect size. The fraction of significant channels does not give one a sense of the magnitude of the improvement both because significance does not reflect effect size and MEG channels are highly correlated, so it is not particularly impressive when all channels pass significance. For example, the authors could plot the correlation between the measured and predicted response (as in the appendix scatter plots) for their base LM and a control model such as GloVE. They could then first test whether their LM showed improved predictions compared with GloVE, which would be a good sanity check, and then could compare the improvement from their fine-tuned model with this increment. I would also find it interesting to evaluate the performance improvement from fine-tuning with the differences between various LM models. Could you get a similar improvement by using a slightly larger vs. smaller model or just training the LM model on a bigger language dataset?

The analyses used to motivate the fine-tuning metric is not compelling. None of the descriptions are significant (which the authors acknowledge) and the procedures used to establish significance are not described. Even if they were significant it is not clear how reliable the effects are. For example, if you ran the procedure twice on different subjects would you get the same thing? Moreover, the labels the authors give feel arbitrary and do not obviously (to me) capture the consensus from the proposed results. I am not confident for example that if you showed this list to two sets of cognitive scientists that they would reliably yield the same categories (i.e., emotional, physical, figurative).

Moreover, the authors show that performance improves for 2 of the 3 tasks they select with weak improvements for one task. Is there any reason to believe that this is better than what you would do if you had simply skipped the whole initial procedure and just tried to come up with any three cognitively relevant and distinct tasks? I am not convinced that the data-driven approach, while interesting, adds value here.

There is no investigation of what it is about the emotion task that yields the best improvements. For example, the emotion task has more training examples than the other tasks. Perhaps, the emotion task is simply more difficult in some sense than the other tasks and training the network on harder tasks yields bigger improvements? There is also no evaluation of whether the fine-tuned models perform well on their task on a different dataset from the one they are fine-tuned on. If they do not generalize well to another dataset for the same task, it would not be surprising if they do not generalize well to neural predictions.

I am assuming for Figure 5 that the authors are reporting MSE improvement for the corresponding model, i.e. when comparing emotion words with non-emotional words, they are using the emotion-trained model. Please clarify this. Assuming this is the case, it would be helpful to know if the effect is specific to the training task, i.e. do you also see a boost for the emotional words when you train on the figurative task, and do you see a boost for figurative words when you train on the emotional task. Please also clarify whether there was correction for multiple comparisons in this figure. In addition, in panel C of Figure 5, the authors seem to report MSE improvements for the physical task that are at least as large if not larger than those in the other two panels, which seems to conflict with the findings from Figure 4. Please clarify/resolve this apparent discrepancy.

It is not clear why they did not select the top 100 sentences that were best and worst predicted, averaging across words. This was the unit of analysis for the proposer-verifier system and so it would be good to know that the sentences were indeed predicted well and badly, respectively.

I think it would be helpful to have a slightly longer (i.e., more than one sentence) description of the propser-validator model. From the description, it is not clear how the hypotheses were validated during trained.

It would be nice if the authors could list all of the emotional and non-emotional words somewhere (same for the other two categories). I understand if that is not feasible, but I think that might provide an easier way to validate whether the labels are good. The Krippendorff’s alpha values seem modest, which made me wonder whether they are high quality.

**Questions:**

Please clarify what dimensions the correlation is being calculated across for Figure 7 through 9 (i.e., time, words, both).

Other questions are described in the weakness section.

---

> ### Author Response · Authors · 2023-11-22
>
> Thanks for the valuable feedback. Please first refer to our official comment for some major changes made to the manuscript.
>
> > I don’t think the statistical procedure used to establish significance is sufficient and it is not described in enough detail to evaluate it. In Appendix F, the authors say they permute the “data” using “random permutation”. What is the data matrix? Are they computing timepoints? Computing timepoints is not valid as this destroys the temporal structure of the data. In addition, the null hypothesis being tested is that there is no relationship between the response and either of the two predictors, which is not the correct null. The correct null is that there is no difference in prediction between the two predictors (but both predictors explain real variance nonetheless).
>
> Thanks for the comment. Just to clarify, the null hypothesis we tested is indeed there is no difference in prediction between the base model and the finetuned model. In order to do that, we compute the empirical p-value on each channel and for each time window. The random permutation is performed across words rather than time points. And because we have (n_channel x n_time) hypotheses tested at the same time, we used FDR to correct p-values. Thanks for pointing out that the writing is ambiguous, we have added details to better describe the statistical procedure.
>
> > In the main figure of the paper, the authors should report a meaningful measure of effect size. The fraction of significant channels does not give one a sense of the magnitude of the improvement both because significance does not reflect effect size and MEG channels are highly correlated, so it is not particularly impressive when all channels pass significance.
>
> Thank you for the comment, a detailed view of the comparison between the base model and the finetuned model for each channel can be found in Appendix H. The effect size can be inferred by the distance of each point from the diagonal line. By observing it one can easily tell that the improvement is significant.
>
> We agree that it would be more rigorous to report effect size in number, but our dumped file only contains correlations rather than individual sample data, so we have to run the whole permutation pipeline to compute effect sizes. Due to the limited time, we cannot add this to the paper but will add it later.
>
> > Moreover, the authors show that performance improves for 2 of the 3 tasks they select with weak improvements for one task. Is there any reason to believe that this is better than what you would do if you had simply skipped the whole initial procedure and just tried to come up with any three cognitively relevant and distinct tasks? I am not convinced that the data-driven approach, while interesting, adds value here.
>
> While we can generate hypotheses on our own, such an approach might overlook certain nuances that are only evident through data-driven method. We believe that, in the long run, a data-driven methodology is more effective than relying solely on intuition, and we want to take a step in that direction. We find it to be a positive sign that the hypotheses generated this way match previous literature on capabilities that may be hard for LMs to acquire through only textual data, but note that it may not always be easy to characterize places where LM and brain responses differ on diverse texts, unless a data-informed method is used.

---

> ### Author Response · Authors · 2023-11-22
>
> > I am assuming for Figure 5 that the authors are reporting MSE improvement for the corresponding model, i.e. when comparing emotion words with non-emotional words, they are using the emotion-trained model. Please clarify this. Assuming this is the case, it would be helpful to know if the effect is specific to the training task, i.e. do you also see a boost for the emotional words when you train on the figurative task, and do you see a boost for figurative words when you train on the emotional task. Please also clarify whether there was correction for multiple comparisons in this figure.
>
> Yes you are right that we reported MSE improvement for models finetuned on corresponding tasks. Indeed, assessing each model's performance on tasks other than the ones they were fine-tuned for would be valuable. However, due to time constraints, we haven't included such an analysis in the current paper but plan to incorporate it in future updates. Regarding your query about multiple comparisons corrections, we initially overlooked this aspect. We have now addressed this and included corrections for multiple comparisons in the revised version of the paper.
>
> > It is not clear why they did not select the top 100 sentences that were best and worst predicted, averaging across words. This was the unit of analysis for the proposer-verifier system and so it would be good to know that the sentences were indeed predicted well and badly, respectively.
>
> Thanks for the suggestion, this is an interesting alternative to explore. However, we choose sentences encompassing best and worst words instead of sentences with averaged best or worst MSEs because averaging could mask sentences with a mix of very high and very low MSEs, making them appear moderate. The goal was to identify “outliers” with notably bad predictions, which wouldn't be as evident through averaging.
>
> > I think it would be helpful to have a slightly longer (i.e., more than one sentence) description of the propser-validator model.
>
> Thanks for the comments, we have added more details about the proposer-validator model to the paper in section 3.1.
>
> > It would be nice if the authors could list all of the emotional and non-emotional words somewhere (same for the other two categories).
>
> Thank you, we do provide some samples of words of each category in Appendix E. A comprehensive list of annotations of words (in csv format) can be found in our repo: https://anonymous.4open.science/r/divergence_MEG-F647.
>
> ## Question:
>
> > Please clarify what dimensions the correlation is being calculated across for Figure 7 through 9 (i.e., time, words, both).
>
> We compute the correlation for each channel and for each time window. The correlation is calculated across words.

---

### Official Review · Reviewer_atcy · 2023-11-01

**Soundness:** 1 poor
**Presentation:** 3 good
**Contribution:** 2 fair
**Rating:** 3
**Confidence:** 3

**Summary:**

The same text sequences are fed at the same time into an LLM and shown to a human test subject. LLM's representations of these text stimuli are captures into an embedding. Human's representations to the stimuli are captured by an MEG device. Then, the representations between humans and LLMs are compared by trying to predict human MEG responses from LLM embeddings.

The analysis focuses on what were the properties of the text that led to lowest prediction scores. From this the authors hypothesize which the linguistic phenomena are the most different between humans and LLMs, pointing at the differences in how languages is processed by these two systems.

**Strengths:**

I like the freshness of the ideas in this paper, however (see below) some of them are too fresh. But I still would like to note originality of the experimental paradigm and the questions posed in this work.

The idea of comparing representations and looking for differences and then figuring out what were the common properties leading to the most different representations is insightful and very interesting.

**Weaknesses:**

MEG signal is very noisy and it captures not only the activity associated with the task (reading the text), but also almost all of the cortical activity that happened to happen at the time. For the rest of the analysis of this paper it is very important that MEG responses we are predicting are actually responses to the text stimuli, but, knowing MEG, I would say this is very unlikely the case. So while you are predicting something from LM representations, it is very unlikely that what you are predicting are responses to text stimuli. Please let me know if MEG source separation or some other new technique has made this possible and I am wrong on this point. But here is a simple experiment you can use to see where the problem is: out of 9,651 words you have recorded responses to, how many of those words you can identify from the MEG signal alone? By just training a decoder from MEG to words. Most probably not many, if any. So how do we know that the MEG signal you are trying to predict contains _any_ signal produced by text processing by the brain?

Figure 1: "LLM-based hypothesis proposer is employed to formulate natural language hypotheses explaining the divergence" -- this sentence is a bit too vague for a scientific paper, the role and function of the Proposer remains unclear after reading such explanation.

Page 2, Contribution 2: How do we know that the "explanations" provided by such a model are even true? Obviously it will generate _an_ explanation, and it would sound plausible as most of the LLM-generated texts do, but how is this approach a method of scientific discovery? How do we replicate / falsify / validate these "hypotheses" that the Proposer is generating? They might be complete rubbish, how do we know they are not?

Page 2, Contribution 3: Too far-reaching conclusions. The experiments provided in this work do not do enough to prove these claims to be scientific discoveries.

Figure 2: How much of this correlation can be attributed to the fact that the regression models were trained on very similar (MEG) data? In order to know if those correlations of up to 0.45 are meaningful one needs to conduct an ablation test, where instead of actual LM representations you feed into your regression models just some random noise. Or to compute the correlations with MEG activity that you know is not the right activity for the presented stimuli. My suspicion is that due to the structure of the data that was modelled the correlations would still be quite high.

Page 3, Section 2.3: "As shown in Figure 2, we observe a temporal pro- gression of accurately predicted areas after word onset" -- What you see is not necessarily a response to the text that the subject is reading, it might be just a response to a visual stimulus appearing in front of their eyes, or some other stimulus-related brain process that does not necessarily contain any signal generated by the neurons that capture the word representation in the subject's brain.

Page 3, Section 3.1: This idea is novel, but relies on too many assumptions and simplifications to be taken seriously as a tool for scientific discovery. The responses given by such an LLM will always contain _some_ analysis, no matter which pairs of D0 and D1 corpora you give it, you will _always_ be able to pick Top 10 reasons for differences. But it does not mean that those differences are actual, valid, or have any relation to the MEG signal. And there was not explanation provided why we should think that they are or do. What if you measure the same set of subjects on the same task on a different day, and/or use a different numpy random seed for your analysis pipeline - would the same top 10 (and top 3) differences come to the top? What if you use another chapter of the book? Would the top 3 differences still be consistent?

**Questions:**

Figure 1: About the ridge regression from LM embedding to discretized MEG signal -- is there one model per time & channel point? Are there (time points) x (channels) models in total?

Figure 1: How is the text input presented to the subject? Do they read the text or listen to it? Is it a long text or just one word?

Page 3, Section 2.2: How well was your custom 1.5B-parameter model able to generate coherent text? This number of parameters is quite small compared to modern models, and GPT-2 abilities are significantly lower that of the most recent open-sourced LLMs.

Page 3, Section 2.2: "For split i, we set aside one fold as the test set" -- How different were the chunks of the signal that were used for the test set from those used for training? Were they just the next time windows, or were they from the same subject, but 10-20-x words away from the training data, or were the test chunks from different subjects?

Were regression models re-trained separately for each subject?

Page 3, Section 2.3: We need some sort of a test here that will convince the reader that all these patterns of activity that you are describing are there because of language and semantic processing. What happens, for example, if instead of actual words we will present test subject with pseudowords, like "feiolg", "dufamping", etc. Or even just just sets of characters like "jkxio", "erkevsd" etc. I have a strong suspicion that even in this case we will be able to observe very similar progression in terms of how MEG activity travels along the brain areas.

Page 5, Table 1: In this analysis were all the 8 subject taken together as one large pool? What if you run the analysis on only 4, and then repeat the analysis on the other 4 -- would the ranking of hypothesis stay the same, would the selected phenomena be the same?

Page 5, Table 2: What if you do this analysis separately per-subject, will they all more or less confirm to the similar ranking of hypotheses or each person would have their own ranking?

---

> ### Author Response · Authors · 2023-11-22
>
> Hi, thanks for your valuable feedback. Please first refer to our official comment for some major changes made to the manuscript.
>
> Regarding your question about whether we can distinguish MEG responses to text stimuli from overall brain activity, our methodology (encoding model) is designed to specifically target activity linked to word processing, effectively filtering out irrelevant activity or noise without the need for source separation. Here's an explanation of our approach:
>
> We first organize the data such that we always look at one time window for all words (e.g. 300-325ms after word onset, for all words). All words take the same amount of time on the screen (500ms). Then we construct a predictive model focusing on this 300-325ms time window. The model is trained to recognize activity directly associated with word processing. Irrelevant activities, such as background noise or unrelated thoughts, are not learned by the model since they do not correlate with the words. These form the basis of encoding models (adjusted for MEG time windows) and are widely recognized for capturing the effects of stimuli [1]. We acknowledge that some word properties the model picks up are visual (as discussed in section 2.3 in the paper), but these are word-specific characteristics (e.g., differentiating between long and short words).
>
> Additionally, our encoding approach can be used to develop a decoder. People have shown that words can be decoded from MEG activity [2], including using this very dataset that we use [3], [4], underscoring that the encoding model captures word-level information. Notably, in [3], word classification is conducted among words of the same length, so the effect of visual features is minimized and it is very likely that the classification is relying on the meaning of the words.
>
> Furthermore, the encoding model was trained on one split of the dataset and tested on another. This is also a common and valid setup in encoding models. The ablation test you propose can be run by randomizing the vectors of the words (in blocks of 20 words, to maintain the potential temporal correlation structure) with respect to the MEG data, and running the experiment multiple times. By experiment we found that the correlation is reliably very close to 0 across channels, which is different from what we observed in the paper.
>
> Regarding your question about the validity of the hypothesis proposer, we propose to validate it by running the model on Chapter 9 and Chapter 10 separately. Notably, the topics identified in Chapter 10 showed a slight variation yet maintained a resemblance to those discovered in Chapter 9.
>
> Needless to say, we have made the necessary corrections to the texts as per your suggestions to ensure they are more rigorous.
>
> [1] Naselaris, Thomas, et al. "Encoding and decoding in fMRI." Neuroimage 56.2 (2011): 400-410.
>
> [2] Sudre, Gustavo, et al. "Tracking neural coding of perceptual and semantic features of concrete nouns." NeuroImage 62.1 (2012): 451-463.
>
> [3] Wehbe, Leila, et al. "Aligning context-based statistical models of language with brain activity during reading." Proceedings of the 2014 Conference on Empirical Methods in Natural Language Processing (EMNLP). 2014.
>
> [4] Toneva, Mariya, and Leila Wehbe. "Interpreting and improving natural-language processing (in machines) with natural language-processing (in the brain)." Advances in neural information processing systems 32 (2019).
>
> [5] Oh, Byung-Doh, and William Schuler. "Why does surprisal from larger transformer-based language models provide a poorer fit to human reading times?." Transactions of the Association for Computational Linguistics 11 (2023): 336-350.

---

> ### Author Response · Authors · 2023-11-22
>
> ## Answers to questions:
>
> > Figure 1: About the ridge regression from LM embedding to discretized MEG signal -- is there one model per time & channel point? Are there (time points) x (channels) models in total?
>
> We trained one model for each time point across all channels. The goal of the model is to predict the whole brain's response at a specific time point given the LM representation of the encountered word. Therefore we have (time points) of models.
>
> > Figure 1: How is the text input presented to the subject? Do they read the text or listen to it? Is it a long text or just one word?
>
> The text is presented to the subjects one word at a time for a fixed duration of 500ms on a screen.
>
> > Page 3, Section 2.2: How well was your custom 1.5B-parameter model able to generate coherent text? This number of parameters is quite small compared to modern models, and GPT-2 abilities are significantly lower that of the most recent open-sourced LLMs.
>
> As we acknowledged in the conclusion, the choice to use GPT-2 was strategic, primarily due to substantial prior research linking GPT-2 embeddings closely with human brain responses. While it's hypothesized that larger language models might yield similar results, there's also contrasting evidence suggesting these bigger models may not align as effectively with human behavior predictions [5]. The exploration of larger models is certainly on the agenda for future research, but given the scope and focus of the current study, continuing with GPT-2 was deemed more appropriate. We will explore larger models such as Llama-2 in future work, and note that the same methodology can also be used with other autoregressive language models.
>
> Also for your reference, the language modeling ability of the base model and fine-tuned models can be found in Appendix I in the paper.
>
> > Page 3, Section 2.2: "For split i, we set aside one fold as the test set" -- How different were the chunks of the signal that were used for the test set from those used for training? Were they just the next time windows, or were they from the same subject, but 10-20-x words away from the training data, or were the test chunks from different subjects?
>
> The split is based on text chunks. Let’s say we are training a model that predicts the brain response to a word 0-25ms after word onset. We first split all texts into 10 chunks (folds). Each time we pick one fold as the test set and train the model on the remaining 9 folds, and we run the model on the test set to get a prediction of it. We do this 10 times to get predictions on all 10 folds, and this procedure is so-called cross-validation. Then because we train a model for each time point, we repeat this cross-validation procedure for each time window of brain responses.
>
> > Were regression models re-trained separately for each subject?
>
> Based on our experience and as you mentioned, MEG is very noisy. Therefore, in our study, we always use the denoised version of MEG data, which takes advantage of cross-subject correspondence, and combines all 8 subjects together.
>
> > Page 3, Section 2.3: We need some sort of a test here that will convince the reader that all these patterns of activity that you are describing are there because of language and semantic processing. What happens, for example, if instead of actual words we will present test subject with pseudowords, like "feiolg", "dufamping", etc. Or even just just sets of characters like "jkxio", "erkevsd" etc.
>
> Previous work [3] shows that classification is possible even after controlling for word length, indicating that the classification uses the semantic properties of words. Thus, if we repeat the experiments with such pseudo-words, we expect that the model might pick up on some of the initial visual responses, but not the later semantic ones.
>
> > Page 5, Table 1: In this analysis were all the 8 subject taken together as one large pool? What if you run the analysis on only 4, and then repeat the analysis on the other 4 -- would the ranking of hypothesis stay the same, would the selected phenomena be the same?
>
>
> Given that Chapter 10 encompasses data from only four subjects, dividing this dataset in half would limit our analysis to just two subjects at a time, potentially introducing too much noise. Therefore, as previously discussed, we opted for an alternate approach to validate our hypothesis. We performed independent analyses for both Chapter 9 and Chapter 10, and observed consistent results across both chapters.
>
> > Page 5, Table 2: What if you do this analysis separately per-subject, will they all more or less confirm to the similar ranking of hypotheses or each person would have their own ranking?
>
> Conducting the analysis on a per-subject basis would significantly reduce the signal-to-noise ratio because we only have single trials for each subject. Estimating the error in predicting a word from just one MEG trial for each subject is bound to be noisy.

---

### Official Review · Reviewer_khQ4 · 2023-11-02

**Soundness:** 2 fair
**Presentation:** 2 fair
**Contribution:** 2 fair
**Rating:** 3
**Confidence:** 5

**Summary:**

This study aims to explore the differences between human language processing and machine language processing with the help of language model (GPT-2) representations and MEG recordings from story reading. The primary contributions can be summarized as follows: exploration of MEG brain encoding for GPT-2 representations fine-tuned on 3 tasks such as emotional understanding, figurative language processing, and physical commonsense tasks.  This resulted in tasks related to emotion and figurative language showing improved alignment with brain responses.

**Strengths:**

1.	The exploration of task-based language model representations and their alignment with the brain is an emerging research direction. Pretrained model representations contain various sources of information sources, whereas task-based representations focus specifically on task-related details.
2.	Considering that three NLP tasks, the authors' examination of their task-based encoding model's performance on MEG recordings is intriguing.

**Weaknesses:**

1.	The idea of task-based modeling and their alignment with Brain is not new. Recently, several research works utilized task-based representations and shown better brain alignment [1], [2], [3], [4].

[1] Oota et al. 2022, Neural Language Taskonomy: Which NLP Tasks are the most Predictive of fMRI Brain Activity?, NAACL-2022

[2] Sun et al. 2023, Fine-tuned vs. Prompt-tuned Supervised Representations: Which Better Account for Brain Language Representations? IJCAI-2023

[3] Aw et al. 2023, Training language models to summarize narratives improves brain alignment, ICLR-2023

[4] Sun et al. 2023, Tuning In to Neural Encoding: Linking Human Brain and Artificial Supervised Representations of Language, ECAI-2023

2.	The novelty in the paper is limited as it primarily delves into a comparison between three task-based model representations and their brain alignment. Notably, the authors have not discussed or compared with 3 previous works [1], [2] and [3].

3.	There are lot of questions left in the methodology:
* Why do authors consider the last layer representations of GPT-2? All the previous linguistic brain encoding studies reveal that intermediate layer representations are well aligned with brain [5], [6], [7].
* Why do authors consider longer context length i.e. 100? Previous linguistic brain encoding studies reveal that context-length of 10-50 have better brain alignment [5], [8].

[5] Toneva et al. 2019, Interpreting and improving natural-language processing (in machines) with natural language-processing (in the brain), NeurIPS-2019

[6] Jain et al. 2018, Incorporating context into language encoding models for fmri, NeurIPS-2018

[7] Oota et al. 2023, Joint processing of linguistic properties in brains and language models, NeurIPS-2023

[8] Oota et al. 2023, MEG Encoding using Word Context Semantics in Listening Stories, Interspeech-2023

4.	The paper lacks clarity regarding its implications. Given the emphasis on comparing fine-tuned representations with vanilla model representations, the influence of the dataset is crucial. If the authors had fine-tuned the model on other emotional datasets, different findings might have emerged.
5.	Additionally, the authors present these task-based findings specifically related to narrative story reading. It's essential to ascertain whether these findings hold true for the listening modality, for instance, by considering datasets like MEG-MASC story listening.
6.	The focus of the authors is primarily on contextual word representations from the last layer of Language Model Models (LLMs). It would be valuable to explore the representation similarity between pre-trained and fine-tuned layer representations. Moreover, investigating which layers are predominantly affected during the fine-tuning process would add depth to the analysis.
7.	The clarity can be improved:
* providing more explicit details concerning the methodology and experimental procedures.
* Figure 4 is hard to follow. More details are need.
8. Several citations are missing [1], [2], [4], and [8]

**Questions:**

1. Why do authors consider the last layer representations of GPT-2? What about the performance of other layer representations?
2. Why do authors consider longer context length i.e. 100?
3. What happens if authors can tune on 3 tasks at same time?
4. Whether the paper findings hold true for the listening modality, for instance, by considering datasets like MEG-MASC story listening?
5. Which layers are predominantly affected during the fine-tuning process would add depth to the analysis?
6.  Why is the parietal region, particularly the angular gyrus, not exhibiting similar enhancement between 250-375ms, considering its association with high-level semantic processing?

---

> ### Author Response · Authors · 2023-11-22
>
> Hi, thanks for your valuable feedback. Please first refer to our official comment for some major changes made to the manuscript.
>
>
> > The idea of task-based modeling and their alignment with Brain is not new. Recently, several research works utilized task-based representations and shown better brain alignment.
>
> We acknowledge that the idea of aligning task-based models with the brain is not novel in itself. As we pointed out in the paper (“Drawing inspiration from ...”), our experiments in the second part of the paper were inspired by prior work. However, the focus of our paper is to address the systematic differences between LMs' and human brains' responses to written language, and we use the “improving LM-brain alignment via fine-tuning on corresponding tasks” approach to validate the hypotheses we identify in the first part of the paper. Unlike previous work which pre-defines a task, our work focuses on discovering what aspects LMs and the brain differ on through a data driven approach, and based on these findings, we propose related tasks for targeted fine-tuning.
>
> > The novelty in the paper is limited as it primarily delves into a comparison between three task-based model representations and their brain alignment. Notably, the authors have not discussed or compared with 3 previous works.
>
> As noted above and in the manuscript, the main aim of this work is to identify and study differences between brain activity recordings and LM activations and suggest hypotheses for what makes LMs different from the brain. We use the task fine-tuning approach to validate our hypotheses. Our aim was not to claim novelty with respect to the fine-tuning. Acknowledging this oversight, we will incorporate references to the omitted works.
>
> > There are lot of questions left in the methodology:
>
> > Why do authors consider the last layer representations of GPT-2? All the previous linguistic brain encoding studies reveal that intermediate layer representations are well aligned with brain.
>
> > Why do authors consider longer context length i.e. 100? Previous linguistic brain encoding studies reveal that context-length of 10-50 have better brain alignment.
>
> Thank you for your valuable comments. Due to limited time, we cannot run all the analyses but we will include these analyses in our paper and validate whether different layers and context window sizes would show similar patterns.
>
> > The paper lacks clarity regarding its implications. Given the emphasis on comparing fine-tuned representations with vanilla model representations, the influence of the dataset is crucial. If the authors had fine-tuned the model on other emotional datasets, different findings might have emerged.
>
> Thank you for the suggestion. As finetuning on multiple datasets will take time, we will reserve this for future work. We will also include a more thorough analysis of the findings and their generalizability.
>
> > Additionally, the authors present these task-based findings specifically related to narrative story reading. It's essential to ascertain whether these findings hold true for the listening modality, for instance, by considering datasets like MEG-MASC story listening.
>
> This is an interesting direction for future work, and we thank you for suggesting this dataset, but based on recent work in fMRI [1], we don’t expect that semantic representation varies much between the reading and the listening modalities. Thus, we expect that the results have a good chance of being similar.
>
> [1] Deniz, Fatma, et al. "The representation of semantic information across human cerebral cortex during listening versus reading is invariant to stimulus modality." Journal of Neuroscience 39.39 (2019): 7722-7736.
>
> > The focus of the authors is primarily on contextual word representations from the last layer of Language Model Models (LLMs). It would be valuable to explore the representation similarity between pre-trained and fine-tuned layer representations. Moreover, investigating which layers are predominantly affected during the fine-tuning process would add depth to the analysis.
>
> Thank you for the suggestion, we will incorporate it in our work.
>
> > The clarity can be improved:
> > providing more explicit details concerning the methodology and experimental procedures.
> > Figure 4 is hard to follow. More details are need.
>
> Thank you for the comments on the writing, we have added details regarding the two points you mentioned.
>
> > Several citations are missing.
>
> Thank you for providing citations that we missed, we have included them in the new version.

---

> ### Author Response · Authors · 2023-11-22
>
> ## Answer to Questions:
> > Why do authors consider the last layer representations of GPT-2? What about the performance of other layer representations?
>
> Replied above.
>
> > Why do authors consider longer context length i.e. 100?
>
> Replied above.
>
> > What happens if authors can tune on 3 tasks at same time?
>
> This is a very good idea. We hypothesize that this would improve LM-brain alignment and it would be interesting to compare the model finetuned on all three tasks with the models finetuned on each task. We will incorporate this suggestion into the next version of our draft, though we do not have enough time now to run the experiments.
>
> > Whether the paper findings hold true for the listening modality, for instance, by considering datasets like MEG-MASC story listening?
>
> Replied above.
>
> > Which layers are predominantly affected during the fine-tuning process would add depth to the analysis?
>
> Replied above.
>
> > Why is the parietal region, particularly the angular gyrus, not exhibiting similar enhancement between 250-375ms, considering its association with high-level semantic processing?
>
> It’s hard to pinpoint the angular gyrus since we have not done source localization. Most of the sensors do exhibit improvement due to fine-tuning. It’s also unclear that 250-375ms is when the angular gyrus gets involved.

---

### Official Review · Reviewer_u3A8 · 2023-11-03

**Soundness:** 3 good
**Presentation:** 3 good
**Contribution:** 3 good
**Rating:** 6
**Confidence:** 1

**Summary:**

This work explores a critical issue about language models with human neural activity, how LMs and humans acquire and use language. In this paper, it explores systematic differences between human and machine language processing using brain data - Magnetoencephalography (MEG). They found three phenomen that LMs may not capture well, they used a GPT-2 XL model to finetune on the dataset for three phenomena, and found that finetuned LMs shows improved alignment with brain responses.

**Strengths:**

1. It is an interesting to explore the connection and difference between LMs and human brain, and finetunes the LMs on a human brain datasets to show the improved alignment with brain responses on three phenomena.

**Weaknesses:**

So far No. (Sorry, I am not an expert for this topic, though I try my best to read this paper)

**Questions:**

So far No. (Sorry, I am not an expert for this topic, though I try my best to read this paper)

---

### Author Response · Authors · 2023-11-22
**Updates to the manuscript**

We have implemented several significant updates to the manuscript:

1. To validate the hypotheses proposed by the automatic hypothesis proposing method, we conduct separate analyses for Chapter 9 and Chapter 10. This time we use a bigger verifier which gives us hypotheses with better validity and p-values. The findings from Chapter 9 are now included in the main body of the paper, while the results for Chapter 10 have been added to Appendix D. Notably, the topics identified in Chapter 10 showed a slight variation yet maintained a resemblance to those discovered in Chapter 9.

2. We observed that the original physical model showed relatively poor performance in its task. To address this, we conducted a grid search to identify more suitable hyperparameters, aiming to enhance the model's performance. This adjustment helps mitigate the issue of the model not adequately representing its domain, a problem not as pronounced in the other two models with their respective tasks. The manuscript now features updated figures reflecting the improved results from the newly fine-tuned physical model. Notably, this fine-tuned version demonstrates better alignment with brain data, supporting the hypothesis proposer's effectiveness in explaining the divergences between the human brain and the language model.

We also made changes to the following sections:

1. Since we observed an improvement in LM-brain alignment across all three tasks, we revised the conclusion in both the abstract and the contributions section.
2. The data preparation section has been restructured to accurately indicate the use of Chapter 10 for validation purposes.
3. Additional details have been included to elaborate on the proposer-verifier hypothesis proposing system.
4. The caption for Figure 4 has been rewritten for greater clarity.
5. Corrections for multiple comparisons have been applied to Figure 5, leading to a reduction in the number of significant time windows identified.
6. References to several previously omitted works have been added.

---

### Meta-Review · Area_Chair_n2dP · 2023-12-09

**Metareview:**

Reviewers overwhelmingly agree that the submission is not yet ready for publication.

Some critical concerns include:
- A disconnect between the narrative and the prior work. While the authors, and part of the submission, acknowledge that prior work addresses related questions even the abstract states "Despite this, there is little work exploring systematic differences between human and machine language processing using brain data.". Reviewers took issue with this characterization of the state of the art and contribution of the submission. This is similarly reflected in contribution (1).
- Reviewers thought the experiments were incomplete and limited. The arbitrary choice of 100 words and arbitrary analysis of only the last part of the network are as the authors state, a consequence of limited time and resources. Such concerns should be addressed before submission. Similarly for using multiple datasets.
- Reviewers also wanted a stronger takeaway, rather than just a demonstration of differences. Why are these differences meaningful? What can we learn from them? There will always be some differences, and countless methods can find differences between brains and machines. Why does this method tell us something useful?
- The takeaway provided, that language models have problems representing some types of knowledge, seems like a large leap that is unsupported by the experiments as the reviewers pointed out.
- Numerous experimental details were missing, particularly around the statistical methods used. Authors promised to clear any confusion up and more clearly describe the experiment. But such explanations are critical for evaluating the submission and cannot be delayed after the review stage.

Reviewers brought up numerous other points which I encourage the authors to engage with. This could be a valuable submission with clearer procedures, a full set of experiments, and a better justified takeaway.

**Justification For Why Not Higher Score:**

The reviewer concerns above. In particular the lack of completeness of both the experiments and the description of the statistical methods. This submission cannot be fully understood and evaluated without these components.

**Justification For Why Not Lower Score:**

N/A

---

### Decision · Program_Chairs · 2024-01-16

Reject